# Back to the Future: Towards Explainable Temporal Reasoning with Large Language Models

## ABSTRACT

Temporal reasoning is a crucial natural language processing (NLP) task, providing a nuanced understanding of time-sensitive contexts within textual data. Although recent advancements in Large Language Models (LLMs) have demonstrated their potential in temporal reasoning, the predominant focus has been on tasks such as temporal expression detection, normalization, and temporal relation extraction. These tasks are primarily designed for the extraction of direct and past temporal cues from given contexts and to engage in simple reasoning processes. A significant gap remains when considering complex reasoning tasks such as event forecasting, which requires multi-step temporal reasoning on events and prediction on the future timestamp. Another notable limitation of existing methods is their incapability to illustrate their reasoning process for explaining their prediction, hindering explainability. In this paper, we introduce the first task of explainable temporal reasoning, to predict an event's occurrence at a future timestamp based on context which requires multiple reasoning over multiple events, and subsequently provide a clear explanation for their prediction. Our task offers a comprehensive evaluation of both the LLMs' complex temporal reasoning ability, the future event prediction ability, and explainability—a critical attribute for AI applications. To support this task, we present the first instruction-tuning dataset of explainable temporal reasoning (ExpTime) with 26k derived from the temporal knowledge graph datasets, using a novel knowledge-graph-instructed-generation strategy. Based on the dataset, we propose the first open-source LLM series TimeLlaMA based on the foundation LLM LlaMA2, with the ability of instruction following for explainable temporal reasoning. We compare the performance of our method and a variety of LLMs, where our method achieves the state-of-the-art performance of temporal prediction and explanation generation. We also explore the impact of instruction tuning and different training sizes of instruction-tuning data, highlighting LLM's capabilities and limitations in complex temporal prediction and explanation generation.

## CCS CONCEPTS

• **Computing methodologies** → **Temporal reasoning**.

## KEYWORDS

Temporal Reasoning, Large Language Models, Event Forecasting, Explainable AI

**ACM Reference Format:**
Anonymous Author(s). 2018. Back to the Future: Towards Explainable Temporal Reasoning with Large Language Models. In *Proceedings of Make sure to enter the correct conference title from your rights confirmation emai (Conference acronym 'XX)*. ACM, New York, NY, USA, 13 pages. https://doi.org/XXXXXXX.XXXXXXX

## 1 INTRODUCTION

Temporal reasoning is a crucial research area in natural language processing (NLP), referring to a model's capability of accurately understand, represent, and predict time-sensitive contexts [13, 31, 33, 54]. This ability is critical for many web-based applications today that rely on processing time-sensitive data, including news article aggregation [21], E-commerce services [49], and search engine recommendation [39]. Existing studies have focused on tasks of temporal relation extraction: predicting the temporal ordering of events [12, 38], temporal knowledge graph (KG) reasoning: inferring missing facts at past and future timestamps [31, 33], and temporal question answering (QA): answering questions requiring multiple steps of temporal relational reasoning [19, 20]. In recent years, the performance of these tasks has been greatly improved by advanced NLP and machine learning methods including graph neural networks (GNNs) [17, 24, 33, 46] and pre-trained language models (PLMs) [34, 45, 57]. Latest, large language models (LLMs) have shown remarkable abilities to understand natural language and human-like text generation [2]. Compared with PLMs, LLMs have a significantly larger model size and pre-training data, and thus show the emergent ability of in-context learning, enabling them to perform unseen tasks without task-specific training data [59]. Inspired by the great promise of LLMs, recent studies have explored the temporal reasoning ability of LLMs [5, 30, 51, 63].

Although these methods have explored the potential and limitations of LLMs, two significant challenges remain conspicuous as shown in Table. 1. Firstly, current methods [5, 30, 51, 63] mainly focused on tasks such as temporal expression detection, normalization, and temporal relation extraction, which are primarily designed for the extraction of direct and historical temporal cues from given contexts and to engage in simple reasoning processes. Therefore, it is still unclear the potential of LLMs for more challenging tasks such as event forecasting [67], which requires multi-step temporal reasoning and the prediction of future timestamps. Secondly, even as LLMs exhibit emergent abilities for in-context learning [59], the area of explainable temporal reasoning – which involves predicting future events from context and explaining the associated reasoning – remains underexplored. It is crucial for models to not only make predictions but also to clearly justify their decisions, to improve transparency. In light of these gaps, we posit the following

| | Explanation | Event Forecasting | Model | Multi-hop Reasoning | Instruction Finetuning | Context Infer |
|---|---|---|---|---|---|---|
| TEMPLAMA [11] | ✗ | ✗ | T5 | ✗ | ✗ | ✗ |
| TEMPREASON [51] | ✗ | ✗ | T5 | ✗ | ✗ | ✓ |
| AutoCast [67] | ✗ | ✓ | T5 | ✓ | ✗ | ✗ |
| ExpTime | ✓ | ✓ | Llama2-7b/13b | ✓ | ✓ | ✓ |

Table 1: The comparison between temporal reasoning datasets and corresponding finetuned models. "Context Infer" denotes if inference based on context is required and "multi-hop reasoning" means engaging in multi-step reasoning is required to arrive at the correct answer.

research questions (RQ) to guide our study: 1) **RQ 1**: Can LLMs be effective in predicting future events by considering the context's complex relations among events, and how do they compare with traditional methods? 2) **RQ 2**: What impact does instruction tuning have, particularly when using our new dataset derived from temporal knowledge graphs, on the temporal prediction capabilities of LLMs? 3) **RQ 3**: How effectively can LLMs clarify their prediction and reasoning process, thereby enhancing their transparency in temporal reasoning tasks?

To address these challenges, our study aims to explore LLMs' capabilities in complex temporal reasoning, future event prediction, and, importantly, explainability—an essential aspect of AI applications. We propose the pioneering task of explainable temporal reasoning, aiming to predict the occurrence of future events based on context, demanding reasoning across multiple events, and subsequently, providing a coherent explanation for the prediction. To support this task, we propose the first-of-its-kind multi-source instruction tuning dataset ExpTime, fostering improvement and assessment of LLMs. ExpTime comprises 26k entries, built from a variety of event forecasting datasets and their derived temporal reasoning paths.

Our methodology begins with aggregating data from various recognized datasets, encompassing diverse sources. For each data point, explanations are generated, drawing inspiration from the proven self-instruct approach [58]. However, we observed that merely prompting LLMs, such as ChatGPT [41], yielded suboptimal results in terms of coherence and accuracy. Recognizing this limitation, we pivoted to a novel Temporal Knowledge Graph-Instructed Generation (GIG) approach. We extract explainable reasoning paths and context from the temporal knowledge graph for each dataset's future event prediction query. We then design prompts to guide LLMs to convert these paths and contexts into coherent explanations. This results in triples of <query, context, answer>, with each answer containing the original prediction and LLM-generated explanation. To ensure the reliability of the dataset, the human evaluation is performed on a subset of the collected data with a carefully designed annotation scheme, evaluating their correctness, completeness, and fluency. We then build a golden-standard testing dataset with human annotation.

Using ExpTime, we propose the TimeLlaMA series, an innovative open-source LLM ensemble based on the LlaMA2 [53], using instruction fine-tuning. Specifically, we fine-tune four TimeLlaMA models: TimeLlaMA-7B, ChatTimeLlaMA-7B, TimeLlaMA-13B, and ChatTimeLlaMA-13B. Our empirical results compare the TemporaLLaMA with other LLMs, highlighting its superior performance

in terms of temporal prediction and explanation generation. Our experiments demonstrate that with proper instruction tuning using even a small volume of high-quality data, the temporal reasoning capabilities of LLMs can be substantially improved. Model size does not necessarily correlate with performance gains in temporal reasoning when employing instruction tuning under 13 billion parameters.

To encapsulate, our contributions are manifold: 1) We pioneer the first task of explainable temporal reasoning, setting the stage for subsequent research, 2) We introduce ExpTime, the first instruction-tuning dataset to improve and evaluate LLMs' ability of explainable temporal reasoning, 3) We propose a novel knowledge graph-instructed generation (GIG) method, for generating explainable temporal reasoning data with LLMs from temporal knowledge graphs, 4) We propose TimeLlaMA, an open-source LLM series tailored for this specific task, achieves SOTA performance, 5) We conduct a holistic evaluation of our method and various LLMs in the realm of temporal reasoning, critically analyze the strengths and limitations of LLMs, providing directions for future research [1].

## 2 RELATED WORK

### 2.1 Temporal Reasoning in NLP

Based on the level of difficulty, temporal reasoning in NLP can be categorized into three tasks: temporal expression detection and normalization, temporal relation extraction, and event forecasting. The temporal expression detection task aims to detect the phrases in the text that describe the temporal information, such as "yesterday" and "last year" [28]. After the detection, the model is required to normalize the temporal expression into a TimeML standard format, such as "2013-01-06". The temporal expression detection and normalization task was first introduced in TempEval-2 [56], where the most successful models are rule-based, such as SUTime and NavyTime [4, 6]. The normalization task was further improved by incorporating pre-trained embeddings later [25, 60].

When time expressions can be detected, the next level of temporal reasoning is to determine the chronological order of events described in the text, namely temporal relation extraction. The temporal relation extraction task was first introduced in TempEval [55]. Initially, this task was tackled by leveraging the sequential neural networks, such as LSTM and RNN, to detect temporal order [12, 34, 40]. Later, GNN was introduced to better capture

---

[1]We will release our models, datasets, and evaluation metrics to the broader research community.

the dependency explicitly between the events and time expressions [3, 38, 62]. As LLMs become popular, some work also investigated the zero-shot ability of LLM in temporal relation extraction and reported that the zero-shot performance is worse than supervised models [5, 63].

With the acquisition of a chronological order of events, the final level of temporal reasoning is event forecasting. The goal of this task is to determine if a specific event will happen in the future given the context events described in the text [21]. Some work has designed a dataset to train the model [67], in which the model can access the context information through links. However, the exploration of this task is still limited despite the importance of this task.

## 2.2 Temporal Knowledge Graph Event Forecast

There are two settings in the temporal knowledge graph reasoning (TKGR) task: extrapolation and interpolation. Extrapolation focuses on predicting whether events will occur in future timestamps, while interpolation aims to complete the temporal knowledge edge graph within a given timespan [31, 33]. Some works also refer to the extrapolation setting as event forecasting in temporal knowledge graph [16, 37]. A key difference between event forecasting in NLP and TKG is the input format - NLP uses textual context, whereas TKG relies on graph structure. To enhance explainability, some methods for TKGR produce predictions along with validated reasons. The explainable methods can be roughly summarized into three categories: logic rule-based approach, reinforcement learning-based approach, and attention network-based approach. For example, TLogic mines logic rules from temporal knowledge graphs for forecasting [37]. Lin et al. proposed graph and logic encoders to incorporate graph information into rules [36]. In reinforcement learning-based (RL) approaches, Sun et al. used an RL agent to travel on the graph to predict events, explaining the prediction [50]. Similarly, Li et al. found event clusters and then searched them with an RL agent [32]. Some models expand an initial query graph via attention until the query entity is reached, using the subgraphs as explanations [16]. Jung et al. also used an attention GNN to iteratively propagate towards the target entity [23]. As explainable TGKR models provide structural reasoning steps on the graph, we leveraged the RL-based and logic-based models to instruct the LLM explanation generation to construct the ExpTime dataset.

## 2.3 Temporal Reasoning in LLM

As growth took place in pre-trained LLMs, a natural question is if LLM is capable of serving as a temporal knowledge base [11, 65]. The pivotal concept of this task is to understand the context under temporal expression and perform temporal-sensitive reasoning to predict missing entities [7]. Temporal datasets have been developed to evaluate LLM on temporal understanding, like CustomNews which evaluates if LLMs can predict masked entities given timestamps [26]. Dhingra et al. then proposed TEMPLAMA which emphasizes temporal questions as $(h, r, ?, t_1)$ and $(h, r, ?, t_2)$ where the answers differ due to different timestamps [11]. TemporalWiki addresses temporal misalignment in LLMs similarly [18]. Tan et al.

expanded TEMPLAMA's time range and added more time-unrelated questions [51].

Some work also further investigated the capability of LLM in the event forecasting task, which is more challenging than temporal-sensitive learning as it requires a full understanding of time and logic. Zhou et al. constructed an Autocast dataset that consists of question-and-answer pairs about future events [67]. Lee et al. tested the zero-shot event forecasting ability of LLM on a temporal knowledge graph and demonstrated that only through in-context learning, LLMs can achieve comparable performance w.r.t current supervised TKG methods [27]. Similarly, Xu et al. designed various prompts to query LLM for temporal knowledge graph completion task [61]. This ability was further improved by few-shot abductive reasoning over LLM and temporal knowledge graph [47]. However, these studies did not evaluate or improve the textual temporal reasoning skills of LLMs. Additionally, the lack of explainability in these LLMs is concerning given their importance in temporal reasoning tasks. To the best of our knowledge, our proposed ExpTime is the first dataset that evaluates and improves the explainability and textual temporal reasoning ability of LLMs.

## 3 METHOD

The objective of this work is to assess and enhance the complex temporal reasoning capabilities of large language models (LLMs). To accomplish this goal, we propose the explainable event forecasting task for complex temporal reasoning and construct the first dataset of its kind: the Explainable Temporal Event Forecasting (ExpTime) dataset. We benchmark the performance of popular LLMs using this new dataset. We then propose the novel LLM series: TimeLlaMA, by instruction finetuning a series of Llama2 models, with the aim of improving the temporal reasoning abilities of LLMs.

## 3.1 Task Definition

We define the explainable temporal reasoning task as follows: given an input document $D$ describing events $\mathcal{E}_{t_1 \sim t_2} = \{e_1, e_2, \cdots, e_n\}$ occurring during time interval $t_1 \sim t_2$, the task is to predict the probability $P = p(e_k | \mathcal{E}_{t_1 \sim t_2})$ that event $e_k$ will occur at future time $t_3$, where $t_3 > t_2 \geq t_1$. Additionally, the LLM must also generate an explanation $F$ that demonstrates its reasoning for the prediction. Each training instance $\mathcal{T}r_i$ for fine-tuning the language model consists of the input document $D_i$, question $Q_i$, prediction answer $P_i$, and explanation $F_i$: $\mathcal{T}r_i = \{D_i, Q_i, P_i, F_i\}$.

## 3.2 Graph-Instruct-Generation: Construct ExpTime Dataset

Recent work has explored using LLMs like ChatGPT to generate datasets by prompting the model to produce answers [48]. However, directly prompting LLMs to generate temporal reasoning data results in low-quality explanations, as we demonstrate in Section 4.2.2. To address this issue, we propose a novel framework called Temporal Knowledge Graph-instructed Generation (GIG) to produce more coherent and accurate reasoning explanations.

The key insight behind our approach is to leverage temporal knowledge graphs (TKGs), which have been effectively utilized for explainable event forecasting. As illustrated in Figure 1, we first apply explainable TKG reasoning models to generate reasoning

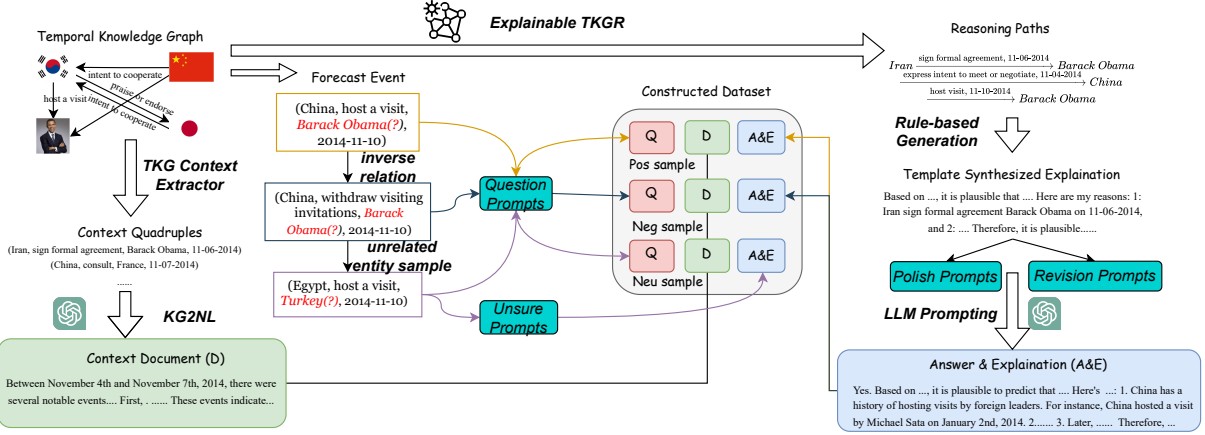

**Figure 1: The pipeline of generating ExpTime dataset. The pos, neg, and neu denote the positive sample, negative sample, and neutral sample, respectively.**

paths for a given query about a future event. We then convert these paths into natural language explanations $F_i$ using a two-level prompting technique we developed. Next, we identify relevant context quadruples from the TKG and reasoning paths to construct a context quadruple set, which is transformed into a coherent natural language document $D_i$. Finally, we convert the original query into a question $Q_i$ to produce a complete training instance $\mathcal{T}r_i = \{D_i, F_i, P_i, Q_i\}$. In this way, our GIG framework overcomes the limitations of directly prompting LLMs by leveraging structured knowledge in TKGs to generate high-quality explanations. The technical details of each step are provided in the following sections.

**Reasoning Paths Generation.** As discussed in Section 2.2, temporal knowledge graph reasoning models can be categorized into three main types. In this work, we select two popular methods representing the most common approaches: TimeTraveler [50], which uses a reinforcement learning-based approach, and TLogic [37], which employs logic rules. We chose these models because they provide high quality and human-readable reasoning chains, as shown in the following equation:

$$(E_1, R_c, E_{m+1}, T_{m+1}) \leftarrow \wedge_{i=1}^{m} (E_i, R_i, E_{i+1}, T_i) \quad (1)$$

where $E_i$, $T_i$, and $R_i$ are the i-th entity, timestamp, and relation, respectively. For example, Fig. 1 shows that given the query quadruple, the explainable TKGR model generates the following reasoning path:

$$Iran \xrightarrow{\text{sign formal agreement, 11-06-2014}} Barack\ Obama$$
$$\xrightarrow{\text{express intent to meet or negotiate, 11-04-2014}} China \quad (2)$$
$$\xrightarrow{\text{host visit, 11-10-2014}} Barack\ Obama$$

To leverage the reasoning chains from these models, we take the average confidence scores (or probability values) of the predictions from the two models and select the reasoning paths $Pa$ with the highest confidence.

**Context Document Generation.** Given a query quadruple $q_u = (e_1, r, e_2, t_i)$, we first extract relevant quadruples from the

TKG to form the context quadruple set, and then transform them into natural language sentences. Specifically, to extract relevant information, we obtain quadruples $q$ that meet two criteria: 1) either entity $e_1$ or $e_2$ from the original query is present in $q$, and 2) the occurrence time $t_q$ of $q$ falls within a defined time span from the query time $t_i$ to time $t_j$. Formally, we extract quadruples $q$ where $(e_1 \in q \vee e_2 \in q) \wedge (t_q > t_i \wedge t_q < t_j)$. We also add the quadruples along the reasoning path $Pa_i$ to the context set.

Once we have the context quadruple set $Q$, the next step is to convert $Q$ into natural language sentences. Prior work such as KELM [1] and GAP [9] have proposed rule-based or pipeline methods, but these cannot generate sufficiently diverse documents from knowledge graphs. Therefore, we designed a prompt to leverage the generative capabilities of ChatGPT to produce more diverse and coherent context documents from $Q$. The prompt is defined as follows:

> *Please generate a coherent paragraph to describe the following quadruples and the time should be precise to dates: [Q]*

In this way, we use the response from ChatGPT as the input document $D_i$ for each training instance $\mathcal{T}r_i$.

**Explanation Generation.** Recall that for each query quadruple $q_u = (e_1, r, e_2, t_i)$, we have obtained the reasoning path $Pa_i$. First, we automatically generate a template-based explanation $F'_i$ for each query quadruple $q_u = (e_1, r, e_2, t_i)$ using the corresponding reasoning path $Pa_i$ obtained from the above steps. This explanation template aims to concisely describe the prediction and the reasoning steps in natural language:

> *Based on the information provided by the document, it is plausible that $e_1$ will $r$ $e_2$ in $t_i$. Here are my reasons: $Pa_1$, and $Pa_2, \cdots$, therefore, it is plausible that $e_1$ will $r$ $e_2$ in $t_i$.*

We refer to this as the template synthesized explanation $F'_i$.

However, these template-generated explanations $F'_i$ may lack coherence or omit critical reasoning details. To improve the quality of explanations, we implement a two-step chain-of-thought (CoT)

prompting approach using LLMs like ChatGPT. First, we prompt ChatGPT to evaluate the correctness of the template explanation $F'_i$ and provide a brief justification, e.g.:

> Given the text, $F'_i$, please evaluate the correctness of the prediction...

We provide detailed prompt in Appendix B.1 for this and all subsequent prompts. If ChatGPT concludes that the explanation $F'_i$ is correct, we propose a "polish prompt" to ChatGPT:

> Can you make the text more coherent and readable by expanding the explanation of each reasoning step?

However, if ChatGPT determines that the template explanation $F'_i$ contains flawed reasoning leading to an incorrect prediction, we provide a "revision prompt" asking ChatGPT to correct the flaws by considering additional context quadruple information from $Q$:

> Please revise ... You can add information from the following quadruples... [Q]

In this way, the ChatGPT response represents the final, improved explanation $F_i$ for each training instance. This CoT prompting approach allows us to leverage the reasoning and language capabilities of LLMs to enhance the quality of automatically generated explanations.

**Negative and Neutral Samples.** Note that by following the previously introduced steps, we can easily acquire positive training instances, i.e., the prediction is that the event will happen. However, using only positive examples to fine-tune language models can lead to highly skewed and imbalanced training. Therefore, we also propose two methods to generate negative and neutral samples individually.

The negative samples represent counterfactual events that did not occur. For each positive training instance $\mathcal{T}r_i = \{D_i, F_i, P_i\}$, we generate a negative example by replacing the relation $r_i$ in the query quadruple $q_u$ with an opposite relation $r'_i$ such that the meaning of $r'_i$ should be as opposite as possible to the original one. For example, we replaced *(Africa, Host a visit, Rex Tillerson, 2018-03-10)* with *(Africa, withdraw visiting invitations, Rex Tillerson, 2018-03-10)*. The resulting negative example quadruple is $q'_u = (e_1, r', e_2, t_i)$. In this way, as the original event did actually happen, the newly synthesized event should be highly unlikely to happen. We manually designed 546 opposite relations for all 265 relations in the temporal knowledge graph. Details are illustrated in Appendix C.

Then, similar to the explanation generation, we first generate a simple template synthesized explanation and then query ChatGPT if the synthesized explanation is correct or not. The prompt is designed as follows:

> Given the text, "Based on..., we predict that $e_1$ $r'$ $e_2$ will not happen in $t_i$. We could find the following patterns from the text: $Pa_1$, and $Pa_2, \cdots$, therefore, it is plausible that $e_1$ will $r$ $e_2$ in $t_i$.", please evaluate the correctness...

Note that the reasoning path $Pa$ is still the same as the positive sample. Then we can obtain the explanation result based on the ChatGPT decision by following the exact same "Polish Prompt" or "Revision Prompt".

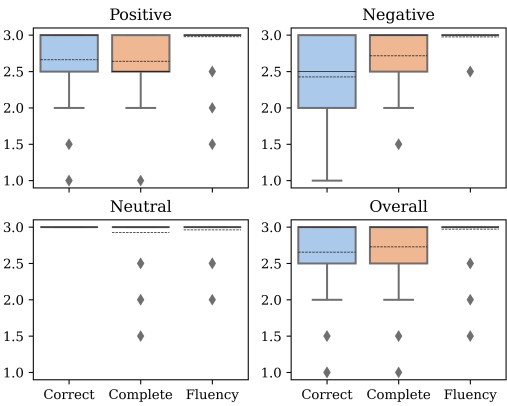

Figure 2: The box plots of human annotation for each criterion under positive, negative, neutral, and overall dataset. The dashed line denotes the mean value and the bold line indicates the median value.

|  | Positive | Negative | Neutral | Overall |
|---|---|---|---|---|
| Correct | 0.73 | 0.64 | 0.81 | 0.74 |
| Complete | 0.65 | 0.59 | 0.70 | 0.66 |
| Fluency | 0.98 | 0.97 | 0.98 | 0.98 |

Table 2: Cohen's Kappa score of human annotation for each criterion under three labels.

In neutral training samples, we expect the LLMs to predict "unsure" for the query quadruple because there is no context information in the given document related to the query. Additionally, for explanation, we also expect the LLMs to summarize the document and then demonstrate that there is no related context in the given document. To achieve this goal, we first replace the query quadruple $q_u = (e_1, r, e_2, t_i)$ with $q''_u = (e'_1, r, e'_2, t_i)$ in the positive training instances, where $e'_1$ and $e'_2$ are entities that never appear in the context quadruple set $Q$. In other words, we ensure the entities in the neutral sample's query do not exist anywhere in the context set $Q$. Formally, we have $e'_1 \notin Q \wedge e'_2 \notin Q$. Then we designed the following prompt to query ChatGPT to generate an explanation:

> Given the document "[$D_i$]", how likely the event that [$e'_1$ $r$ $e'_2$] in [$t_i$] would happen? ... if the context is unrelated, summarize the context...

## 3.3 Data Statistics and Annotation

We utilize ICEWS14 [15], ICEWS18 [22], and ICEWS0515[15] datasets to generate the proposed dataset, as they are the most popular temporal knowledge graph reasoning datasets. From the three datasets, we extracted 12,229 reasoning paths and therefore generated 12,229 positive samples in the dataset. The detailed statistics of ExpTime are shown in Table. 3.

To further evaluate the quality of our dataset and construct a standardized testing dataset, two experienced annotators independently

|       | Pos.  | Neg. | Neu. | ICEWS14 | ICEWS18 | 0515 |
|-------|-------|------|------|---------|---------|------|
| Train | 11703 | 8705 | 5360 | 10327   | 7651    | 7790 |
| Test  | 435   | 300  | 266  | 387     | 351     | 263  |

**Table 3: The statistics of constructed dataset. Pos., Neg., Neu. denote the number of positive, negative, and neutral samples, respectively.**

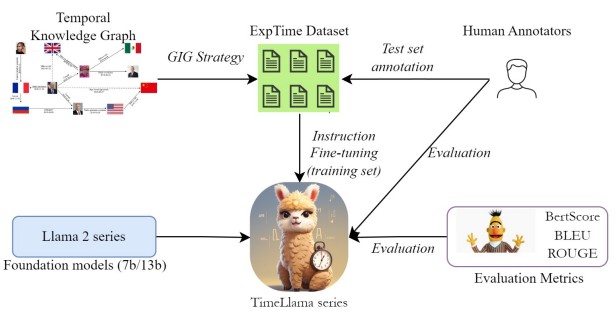

**Figure 3: The pipeline of finetuning and evaluating TimeLlama series models. GIG Strategy denotes our proposed dataset construction approach.**

evaluated a random sample of 1,200 explanations. The annotators rated each explanation on three criteria: 1) correctness, which assessed whether the prediction and explanation were accurate; 2) completeness, which evaluated if the explanation provided the necessary context to understand the prediction; and 3) fluency, which measured if the explanation was clear and understandable. The annotation guidelines and annotator qualifications are detailed in Appendix A. Cohen's kappa coefficient was calculated to determine inter-rater agreement for each criterion. As shown in Table 2, a high level of agreement was achieved for all criteria. In particular, the annotators demonstrated strong agreement on fluency ratings and agreement was higher overall for samples receiving neutral labels. As illustrated in Fig. 2, most samples received high scores across all three criteria. The strong inter-rater agreement and generally high scores indicate the testing dataset represents a high-quality, standardized sample for evaluation. Low-scoring samples on any of the criteria were excluded.

### 3.4 TimeLlama

As illustrated in Fig. 3, we present the TimeLlama model series, representing the first LLMs fine-tuned specifically for complex temporal reasoning tasks, namely explainable event forecasting. By instruction tuning the models on datasets requiring the comprehension and synthesis of temporal information, TimeLlama gains an enhanced ability to make logical inferences about the timing, duration, and relations between events. This supports a more accurate prediction of what events may occur next given a historical context. We construct TimeLlama-7b and TimeLlama-13b by finetuning the base Llama-7b and Llama-13b models, respectively. The finetuning

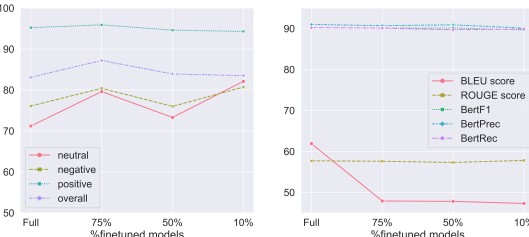

**Figure 4: The automatic evaluation scores of finetuned Llama2 with various percentages of dataset usage. From left to right: F1 scores of each category, BLEU and ROUGE scores, BERTScore.**

process utilizes Flash Attention and DeepSpeed to accelerate training [10, 44]. Full hyperparameters can be found in Appendix D.2. Additionally, by finetuning the Llama-7b/13b conversational models, we construct ChatTimeLlama-7b and ChatTimeLlama-13b, based on Llama-2-Chat-7b/13b optimized using reinforcement learning from human feedback (RLHF) [41].

## 4 EXPERIMENTS

### 4.1 Experimental Settings

**Baselines.** We evaluate and compare the following LLMs as the baselines: **Flan T5** [8]: An instruction-finetuned T5 model based on chain-of-thought data that increased the number of tasks. **BART** [29]: An encoder-decoder architecture model that is proficient in abstractive dialogue, question answering, and summarization tasks. **MPT-7b** [52]: A LLM that is optimized for extremely long inputs. The MPT model with 7b parameters fine-tuned for dialogue generation is used in our experiment. **Falcon-7b** [43]: A LLM that is optimized for faster inference with decoder-only architecture. The 7B dialogue-fine-tuned version is used. **Vicuna-7b** [66]: A chatbot trained by fine-tuning LLaMA on a dataset collected from ShareGPT. **ChatGPT** [14]: A chatbot based on GPT-3.5 LLM that is capable of having natural conversations. **Llama2-7b/13b-chat** [53]: Llama-2 is a collection of open-sourced LLMs that outperform other models in most tasks. The chat-fine-tuned Llama2-7b/13b is used.

**Metrics.** Our evaluation can be roughly divided into automatic and human evaluation. In automatic evaluation, we first report precision, recall, and F1 scores of event predictions. For explanation evaluation, we choose BLEU [42] (unigram, bigram, 3-gram, 4-gram) and ROUGE [35] (rouge1, rouge2, rougeL) to compare the explanation generated by the LLMs with the golden explanations in the testing set. Besides the metric-based methods, we also report the BertScore [64] that computes the similarity based on PLMs. We use the same evaluation criteria introduced in Sec. 3.3 for human evaluation, namely correctness, completeness, and fluency.

### 4.2 Automatic Evaluation Results

*4.2.1 Prediction Evaluation.* In Table 4, we present compelling evidence of the substantial enhancements achieved through the fine-tuning of the ChatTimeLlama-7b model. Notably, our finetuned Llama2-7b model surpasses its baseline counterpart across multiple performance metrics. Specifically, we observe impressive F1 gains

| Models | Positive | | | Negative | | | Neutral | | | Overall | | |
|---|---|---|---|---|---|---|---|---|---|---|---|---|
| | Prec | Recl | F1 | Prec | Recl | F1 | Prec | Recal | F1 | Prec | Recal | F1 |
| Flan T5 | 62.9 | 29.2 | 39.9 | 31.4 | 57.0 | 40.5 | 32.2 | 30.8 | 31.5 | 45.3 | 38.0 | 38.0 |
| BART | 45.7 | 28.2 | 34.9 | 26.3 | 11.7 | 16.2 | 21.5 | 18.3 | 19.8 | 33.6 | 27.0 | 25.3 |
| MPT-7b | 48.5 | 64.6 | 55.4 | 39.5 | 35.7 | 37.5 | 25.8 | 14.7 | 18.7 | 39.8 | 42.7 | 40.3 |
| Falcon-7b | 47.7 | 56.6 | 51.7 | 37.9 | 22.0 | 27.8 | 19.9 | 23.3 | 21.5 | 37.4 | 37.4 | 36.5 |
| Vicuna-7b | 48.4 | 80.5 | 60.4 | 41.3 | 21.3 | 28.1 | 35.8 | 16.5 | 22.6 | 42.7 | 45.6 | 40.4 |
| ChatGPT | 90.9 | 39.1 | 54.7 | 29.5 | 31.7 | 30.5 | 30.7 | 56.8 | 39.8 | 56.5 | 41.6 | 43.5 |
| Llama2-7b-chat | 50.1 | 83.9 | 62.7 | 41.9 | 13.0 | 19.8 | 27.4 | 18.4 | 22.0 | 41.6 | 45.3 | 39.1 |
| Llama2-13b-chat | 51.3 | 53.8 | 52.5 | 40.0 | 26.0 | 31.5 | 28.0 | 36.8 | 31.8 | 41.7 | 41.0 | 40.7 |
| TimeLlama-7b | 90.1 | 97.6 | 93.7 | 67.6 | 84.9 | 75.3 | 97.8 | 55.1 | 70.5 | 84.6 | 82.7 | 81.5 |
| ChatTimeLlama-7b | 91.3 | 99.3 | 95.2 | 68.3 | 86.0 | 76.1 | 98.7 | 55.6 | 71.2 | 86.4 | 83.7 | 83.1 |
| TimeLlama-13b | 94.6 | **100** | 97.2 | 73.9 | 91.3 | 81.7 | **99.4** | 63.5 | 77.5 | 89.6 | 87.7 | 87.3 |
| ChatTimeLlama-13b | **96.2** | 99.5 | **97.9** | **75.0** | **94.0** | **83.4** | 98.9 | **65.0** | **78.5** | **90.6** | **88.7** | **88.4** |

Table 4: The prediction performance of each model on gold temporal reasoning testing set. The overall denotes the weighted average precision, recall, and F1 score.

| Models | BLEU | | | | | ROUGE | | | | BertScore | | |
|---|---|---|---|---|---|---|---|---|---|---|---|---|
| | unigram | bigram | 3-gr | 4-gr | avg | rouge1 | rouge2 | rougeL | rougesum | Prec | Recal | F1 |
| Flan T5 | 48.9 | 16.9 | 9.7 | 6.6 | 15.2 | 28.9 | 7.5 | 26.1 | 26.0 | 77.6 | 78.1 | 76.9 |
| BART | 24.5 | 11.5 | 7.1 | 4.3 | 8.9 | 23.1 | 6.3 | 19.8 | 19.7 | 75.3 | 76.7 | 74.9 |
| MPT-7b | 28.7 | 11.9 | 7.4 | 5.2 | 10.7 | 31.8 | 13.4 | 26.9 | 27.2 | 81.4 | 80.5 | 80.1 |
| Falcon-7b | 54.9 | 22.7 | 13.6 | 9.1 | 19.8 | 33.8 | 13.9 | 29.2 | 29.3 | 80.3 | 80.4 | 79.9 |
| Vicuna-7b | 60.5 | 27.7 | 16.5 | 10.9 | 23.5 | 43.3 | 19.7 | 37.1 | 37.2 | 83.7 | 83.8 | 83.3 |
| ChatGPT | 66.9 | 34.7 | 23.5 | 17.2 | 31.1 | 42.2 | 22.6 | 37.1 | 37.1 | 84.8 | 83.8 | 83.7 |
| Llama2-7b-chat | 61.9 | 30.4 | 19.6 | 13.9 | 26.8 | 44.2 | 23.7 | 38.3 | 38.4 | 84.2 | 84.5 | 83.8 |
| Llama2-13b-chat | 60.9 | 29.3 | 18.5 | 12.9 | 25.5 | 42.4 | 21.8 | 36.6 | 36.6 | 83.7 | 84.1 | 83.4 |
| TimeLlama-7b | 77.5 | 50.5 | 38.8 | 30.7 | 59.9 | 46.3 | 29.6 | 56.6 | 56.5 | **91.0** | **90.2** | **90.2** |
| ChatTimeLlama-7b | **78.3** | **52.4** | **40.2** | **32.6** | **61.9** | 48.2 | **31.1** | **57.6** | **57.7** | 89.2 | 88.3 | 88.8 |
| TimeLlama-13b | 76.5 | 48.8 | 36.5 | 29.1 | 44.6 | 59.4 | 29.5 | 54.9 | 54.9 | 90.0 | 89.4 | 89.4 |
| ChatTimeLlama-13b | 77.4 | 50.5 | 38.2 | 30.7 | 46.3 | **60.7** | 30.2 | 56.2 | 56.3 | 90.5 | 89.7 | 89.7 |

Table 5: The explanation generation performance of each model on gold temporal reasoning testing set.

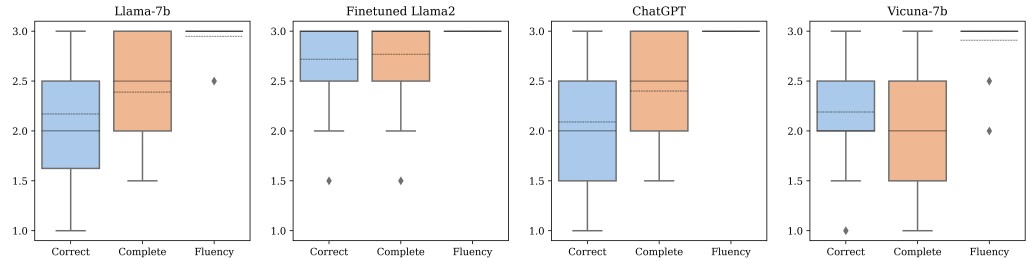

Figure 5: The box plots of human evaluation for each LLM. The dashed line denotes the mean value and the bold line indicates the median value.

improvements of 44.0, 32.5, 56.3, and 49.2 across four categories: positive, negative, neutral, and overall. These figures underscore the efficacy of our fine-tuning approach, even in the presence of noise within the training dataset. Notably, this underscores the capacity of LLMs to leverage high-quality generated datasets by instruction-tuning for substantial performance enhancements.

**Bigger LLM is not always better.** Interestingly, increasing the model scale does not necessarily improve performance. Doubling the parameters from Llama2-7b-chat to Llama2-13b-chat yielded

only marginal gains, with Llama2-7b-chat actually outperforming Llama2-13b-chat on the positive class. For instance, Llama2-7b-chat has a 10.2 F1 gain compared with Llama2-13b-chat in the positive category. Another example is the comparison between MPT-B and Flan T5. For instance, when we examined the "unsure" category, we observed that Flan T5 demonstrated an impressive F1 score of 31.5. It outperforms both MPT-7b and Falcon-7b, which achieved F1 scores of 18.7 and 21.5, respectively.

**ChatGPT performs mediocre in the zero-shot setting.** Notably, even though our dataset is generated by prompting ChatGPT, it is evident that ChatGPT exhibits suboptimal performance when presented with direct prompts, in contrast to our dataset construction approach. To provide a comprehensive view of ChatGPT's performance, we compare it with Vicuna-7b, a model that was not involved in the dataset construction process. The results reveal that ChatGPT achieves an overall F1 score of 43.5, while Vicuna-7b demonstrates a comparable F1 score of 40.4. Furthermore, our fine-tuned model, Llama2-7b-chat, exhibits a substantial 39.6 F1 point improvement over ChatGPT's performance.

*4.2.2 Explanation Evaluation.* In Table 5, we present the automatic evaluation results for the explanation generation. Notably, our fine-tuned variant, ChatTimeLlama-7b, demonstrates remarkable improvements across all key evaluation metrics. For instance, when compared to the baseline Llama2-7b-chat, ChatTimeLlama-7b exhibits substantial enhancements in BLEU, ROUGE, and BertCore scores, with gains of 35.1, 19.3, and 6.4 points, respectively. These results underscore the significant potential for enhancing the explainable temporal reasoning capabilities of LLMs through instruction tuning based on high-quality datasets.

Parallel to our prediction evaluation, our examination of explanation quality yields insightful observations. First, our explanation evaluation results also demonstrate that ChatGPT with direct prompting exhibits limitations in generating coherent reasoning explanations. For example, the BLEU and ROUGE scores of ChatGPT are 31.1 and 37.1 while Llama2-7b-chat can also achieve comparable performance, i.e., 26.8 BLEU score and 38.4 ROUGE score. We include a failure example of ChatGPT in Appendix B.3. Second, the explanation quality of TimeLlama-13b is not better than that of TimeLlama-7b. For example, ChatTimeLlama-7b achieves a 61.9 BLEU score while ChatTimeLlama-13b has 46.3 BLEU. This may be due to overfitting, and lack of grounding where maximizing prediction harms explainability.

Another interesting finding is that even Flan T5 and BART can achieve comparable performance on prediction evaluation, these two LLMs along with MPT-7b produce subpar explanations compared to other LLMs. One possible reason could be the different coverage of their training dataset and the difference between "encoder-decoder" and "decoder" only architecture.

## 4.3 Human Evaluation Results

To provide an objective assessment of the quality of the generated explanations, two experienced annotators evaluated explanations from four language models: Llama2-7b, TimeLlama2-7b, ChatGPT, and Vicuna-7b. The annotation guidelines and annotator qualifications are detailed in Appendix A. 50 explanations from each model were randomly selected, paired with the corresponding question,

and evaluated by the annotators. As shown in Figure 5, the results demonstrate that overall the TimeLlama2-7b model achieved the highest scores across the three assessment criteria. Specifically, all models generated fluent explanations, as indicated by the high fluency scores. Llama2-7b and ChatGPT performed similarly on correctness and completeness. Compared to the baseline Llama2-7b, the TimeLlama2-7b showed significantly improved correctness and completeness, suggesting that finetuning on the high-quality dataset enhanced its ability to provide coherent temporal reasoning explanations. Cohen's kappa coefficients in Appendix A also show a high level of inter-annotator agreement for most model evaluations. In summary, the finetuned Llama2 model generated the highest quality explanations according to the human evaluation, demonstrating the efficacy of finetuning on a curated dataset to improve the explanatory capabilities of language models for temporal reasoning.

## 4.4 Fractional Data Trains LLM Reasoning Skills

Previous experiments have demonstrated that fine-tuning LLMs on high-quality datasets can significantly improve their ability to provide explainable temporal reasoning. This leads to an investigation of the minimum amount of high-quality data required to improve the explainable temporal reasoning capabilities of LLMs. To test this, 10%, 50%, and 75% of the training samples were randomly selected from the dataset to fine-tune Llama2-7b using the same fine-tuning methodology. Interestingly, Llama2 fine-tuned on reduced amounts of data achieved comparable or better performance on automatic prediction and explanation evaluation metrics in some cases (Fig. 4). For instance, Llama2 fine-tuned on 75% of the dataset attained a higher F1 score for prediction accuracy compared to the full dataset. Moreover, the Llama2 fine-tuned on just 10% of the data obtained similar performance on explanation metrics such as ROUGE score and BERT score versus Llama2 fine-tuned on 75% and the full dataset. These results demonstrate that with guidance from even a small volume of high-quality data, the temporal reasoning and explanation generation skills of LLMs can be substantially enhanced.

## 5 CONCLUSION

In this work, we propose the first task of explainable temporal reasoning, to predict an event's occurrence at a future timestamp and generate the explanation for their prediction. To support this task, we introduce a novel dataset ExpTime, containing 26k examples derived from temporal knowledge graphs, developed by a novel knowledge-graph-instructed-generation strategy. Based on this dataset, we develop TimeLlama, an open-source LLM series tuned with instructions for temporal reasoning and explanation generation. Experiments demonstrate the SOTA performance of TimeLlama on future event prediction and explanation generation compared to other LLMs. We find the instruction-tuning using high-quality data is critical for improving LLM's temporal reasoning and explainability. We discuss associated ethical considerations and limitations in Appendix F. In the future, we plan to expand the breadth and diversity of our benchmark dataset by incorporating more temporal reasoning tasks.

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

# A HUMAN ANNOTATION AND EVALUATION

## A.1 Settings

We recruited two Computer Science Ph.D. students with expertise in natural language processing (NLP) to manually annotate the test dataset and evaluate the performance of each model. The annotators were provided with the full sample set for each instance in the test data, consisting of the input document, the question about future events, and the ground truth answer with an explanation. To evaluate model performance, the annotators were also shown the model-generated explanation alongside the complete sample. By having domain experts manually annotate the testing data and compare model outputs, we aimed to robustly assess the ability of each model to provide accurate and logical explanations.

## A.2 Annotation Guideline

Here we describe our human annotation guidelines for annotating and evaluating the prediction and explanation quality.

*Overview:* You will evaluate machine-generated predictions about future events along with explanatory reasoning. The predictions and explanations are based on a given context document. Please rate each answer on a scale of 1 to 3 using the criteria below: *Prediction Accuracy (1-3):*

- 1 - The prediction on whether the event will occur is incorrect. For example, the prediction is that Event X will occur, but Event X does not happen
- 2 - The prediction on whether the event will occur is correct but the reasoning is flawed
- 3 - The prediction and reasoning are fully accurate and aligned

*Explanation Completeness (1-3):*

- 1 - The explanation does not provide the necessary context/background to support the prediction. (e.g. "This will happen because of past events" with no further details)
- 2 - The explanation provides some relevant context but lacks important details

|  | Correctness | Completeness | Fluency |
|---|---|---|---|
| Llama-7b | 0.73 | 0.47 | 0.98 |
| Finetuned Llama2 | 0.82 | 0.54 | 0.99 |
| ChatGPT | 0.77 | 0.41 | 0.98 |
| Vicuna | 0.74 | 0.50 | 0.97 |

**Table 6: Cohen's Kappa score of human evaluation for each criterion in each LLM.**

- 3 - The explanation comprehensively provides the background needed to understand the prediction

*Explanation Fluency (1-3):*

- 1 - The explanation is unclear or difficult to understand, such as wordy, confusing, or unclear connections between ideas, etc.
- 2 - The explanation could be improved stylistically but is reasonably clear
- 3 - The explanation is fluent, coherent, and easy to comprehend

### A.3 Agreement Level

We present the human evaluation agreement level by using Cohen's Kappa score in this section. Note that the completeness inter-agreement level is relatively lower compared with the other two criteria. The main reason is that the completeness score heavily relies on the annotator's own logic and domain knowledge, which may introduce the annotator bias.

## B GIG STRATEGY EXAMPLES

### B.1 Detailed Prompts

*B.1.1 Template-generated explanation evaluation prompt.* Given the text, $F_i'$, please evaluate the correctness of the prediction based on the reasoning steps shown in the text. Answer correct or wrong then explain your decision concisely

*B.1.2 revision Pprompt.* Please revise the provided text to ensure the prediction aligns with the reasoning steps. Adjust the flaws accordingly to reflect a correct prediction. Emphasize the importance of a logical progression of reasoning. You can add information from the following quadruples only if it is necessary for making the correct prediction. Finally, make the whole revised text more readable and coherently by expanding the explanation of each reasoning step. [$Q$]

*B.1.3 negative sample generation prompt.* Given the text, "Based on the information provided by the document, we predict that $e_1$ $r'$ $e_2$ will not happen in $t_i$. We could find the following patterns from the text: $Pa_1$, and $Pa_2, \cdots$, therefore, it is plausible that $e_1$ will $r$ $e_2$ in $t_i$.", please evaluate the correctness of the prediction based on the reasoning steps shown in the text. Answer correct or wrong then explain your decision concisely

*B.1.4 neutral sample generation prompt.* Given the document "[$D_i$]", how likely the event that [$e_1'$ $r$ $e_2'$] in [$t_i$] would happen? Please note that the context shown in the given document may not be

(Iran, sign formal agreement, Barack Obama, 11-06-2014)
(China, consult, France, 11-07-2014)
......
(Barack Obama, express intent to meet, China, 11-04-2014)

**Table 7: The context quadruples for query *(China, host a visit, Barack Obama(?), 2014-11-10)***

directly related to the event, so your answer should be uncertain. And if the context is unrelated, summarize the context and tell me why you think the context is not related.

### B.2 A GIG Workflow Example

To better help readers understand our proposed GIG explainable temporal reasoning dataset construction strategy, we provide a detailed example here to show the workflow of GIG.

First, we identify a query on temporal knowledge graph *(China, host a visit, Barack Obama(?), 2014-11-10)*. Then a reasoning path from explainable temporal knowledge graph reasoning models can be obtained:

$$Iran \xrightarrow{\text{sign formal agreement, 11-06-2014}} Barack\ Obama$$
$$\xrightarrow{\text{express intent to meet or negotiate, 11-04-2014}} China \quad (3)$$
$$\xrightarrow{\text{host visit, 11-10-2014}} Barack\ Obama$$

. We then extract quadruples/events that happened during the past week of 2014-11-10 that include either China or Barack Obama and name them context quadruples. We also include the quadruples on the reasoning path in the context. The context quadruples are shown in Table 7 We then generate a very simple natural sentences-format explanation based on the reasoning path:

> *Based on the information provided by the document, it is plausible that China will host a visit Barack Obama on 11-10-2014. Here are my reasons: 1: Iran sign formal agreement Barack Obama on 11-06-2014, and 2: Barack Obama express intent to meet or negotiate on 11-04-2014 China. Therefore, it is plausible that China will host a visit Barack Obama on 11-10-2014*

We prompt this explanation and the context quadruples to ChatGPT following the polish or revise prompt. Then ChatGPT-generated explanation can be obtained:

> *Yes. Based on the information provided in the document, it is plausible to predict that China will host a visit by Barack Obama on November 10, 2014. Here's a breakdown of the logical progression of reasoning to support this prediction: 1. China has a history of hosting visits by foreign leaders. For instance, China hosted a visit by Michael Sata on January 2nd, 2014. 2. In a reciprocal fashion, following Sata's visit, Michael Sata made a visit to China on the same day of January 2nd, 2014. 3. Later, China also hosted a visit by Barack Obama on March 4th, 2014. Based on these three points, it can be inferred that China has established a pattern of hosting visits by foreign leaders, including both Michael Sata and Barack Obama.*

> *Therefore, it is logical to conclude that China may host a visit by Barack Obama on November 10th, 2014.*

We obtained the answer and explanation in this way.

Then we construct the context document by prompting ChatGPT to generate coherent sentences to describe the context quadruples:

> Between November 4th and November 7th, 2014, there were several notable events related to international relations. First, on November 4th, Barack Obama expressed his intent to meet or negotiate with China. ...... These events indicate a series of significant diplomatic engagements between various countries and key figures during this period.

The input document can be built in this way.

We convert the query into natural language format by injecting the query into a set of diverse question templates:

> Given the following document, can we expect that China will host a visit Barack Obama on 2014-11-09? Can you please answer with a yes, no, or unsure first, and then explain your decision?

### B.3 Direct Prompting Failure of ChatGPT

In order to directly compare the performance of our approach to that of ChatGPT, we have included an example of a ChatGPT failure case in Table 8. As illustrated, ChatGPT struggled to detect the underlying logic and causal relationships between events described in the provided document. In contrast, our proposed approach was able to successfully predict and reason about the sequence of events. This comparison highlights the challenges of relying solely on direct prompting of large language models like ChatGPT for dataset construction. Our results suggest that a more structured, programmatic approach to dataset creation, like the one proposed in the paper, may be necessary to produce high-quality training data with reliable logical reasoning abilities.

### C REVERSE RELATIONS

As described in the section on negative sample generation, we carefully manually designed inverse relations for all relations present in the original temporal knowledge graph. This ensures that if an event occurs, the inverse event is highly unlikely to occur. Table 9 provides examples of the inverse relations we crafted.

### D EXPERIMENTS BACKGROUND

#### D.1 ICEWS dataset

The ICEWS datasets are built from the Integrated Crisis Early Warning System, which monitors and analyzes world events to identify potential crises. The most popular ICEWS datasets are ICEWS14, ICEWS18, and ICEWS0515. The number denotes the year of the events in each dataset. The statistics of each dataset are shown in Table. 10.

#### D.2 Hyper-parameters

Training hyperparameter details are as follows. A per-device batch size of 2 was utilized with a gradient accumulation step of 16. Optimization was performed with the AdamW algorithm, employing a

peak learning rate of 2e-5 and a warm-up ratio of 0.03. The maximum model input length was set to 2048 tokens. DeepSpeed ZeRO stage 3 was enabled for optimization. All models were trained using 4 Nvidia Tesla A100 GPUs, each with 80GB of memory.

### E UNEXPLAINABLE TEMPORAL KNOWLEDGE GRAPH EVENT FORECASTING MODELS

Here we examine methods developed for the temporal knowledge graph event forecasting task, which aim to improve predictive accuracy rather than explainability. Specifically, we review models designed for this forecasting task that do not incorporate explainable components into their architectures. The goal of these models is to enhance predictive correctness on the forecasting benchmark, without considerations for explainability or interpretability. By focusing solely on improving forecasting performance, these methods provide a baseline to compare future work on building explainable forecasting models.

Specifically, in TKG event forecasting, the first effort is Know-evolve, which captures the continuous-time temporal dynamics and predicts future facts by estimating the conditional probability of temporal point process [54]. With the rise of graph neural networks (GNN), Relational-GCN has been introduced into event forecasting to replace temporal point process [46]. The two influential RGCN-based methods are RE-NET and RE-GCN, where RE-NET utilized RGCN to encode long-term representations of temporal knowledge graphs by designing an autoregressive event recurrent encoder [22] and RE-GCN proposed to focus on the graph dependency structure [33].

### F ETHICAL CONSIDERATIONS AND LIMITATIONS

In developing this temporal reasoning dataset, care has been taken to ensure appropriate consideration of ethical issues and limitations commonly associated with large language models. The source data has been carefully curated to provide diversity and mitigate biases. Events representing a wide range of demographic groups are included to avoid propagating systemic stereotypes. We acknowledge that, despite best efforts, the dataset may exhibit gaps or contain unintended biases. Finally, we recognize that large language models carry risks of generating harmful, biased, or logically incoherent content through hallucination. Our evaluation methodology takes this into account by prioritizing answer accuracy over fluency. With rigorous design and testing processes, we aim to uphold ethical AI principles while furthering research on temporal reasoning.

| | |
|---|---|
| Question | Given the document, "On March 12th, Oman and Iran engaged in consultations. The following day, on March 13th, Media Personnel from Iran made empathetic comments. Additionally, Majid Takht Ravanchi from Iran demanded something on March 13th. In another development, Business (Iran) made a statement regarding Iran on March 10th. Turning to international relations, Iran hosted a visit from Evangelos Venizelos on March 14th. Furthermore, on January 4th, Iran hosted a visit from the Envoy of the United Kingdom, while on the same day the Envoy made a visit to Iran. Lastly, on March 9th, Iran and Iraq signed a formal agreement, and on March 12th, Iran also signed a formal agreement with Iraq.", is it possible that Iran will host a visit Evangelos Venizelos on 2014-03-16? |
| ChatGPT Answer | Based on the information provided in the document, Iran hosted a visit from Evangelos Venizelos on March 14th, 2014. There is no mention of Iran hosting a visit from Evangelos Venizelos on March 16th, 2014, in the document. Therefore, it is not possible that Iran will host a visit from Evangelos Venizelos on that date based on the information provided. |
| Revision Answer | Yes. Based on the information provided by the document, it is plausible that Iran will host a visit from Evangelos Venizelos on 2014-03-16. Here are the reasons to support this prediction: 1. On 2014-01-04, Iran hosted a visit from the United Kingdom envoy. This implies that Iran is open to hosting visits from foreign diplomats or officials. 2. On the same date, the United Kingdom envoy visited Iran. This suggests that the two countries have established diplomatic relations and engage in reciprocal visits. 3. On 2014-03-15, Iran hosted a visit from Evangelos Venizelos. This indicates that Iran is actively involved in diplomatic engagements and willing to host visits from foreign officials. Based on these reasons, it is reasonable to assume that Iran would continue its diplomatic activities by hosting a visit from Evangelos Venizelos on 2014-03-16. |

**Table 8: Comparison between directly prompting ChatGPT and the proposed dataset construction method**

| Relation in TKG | Reversed relation |
|---|---|
| Accede to demands for change in institutions, regime | "Reject request for change in institutions, regime","Reject request or demand for political reform","Reject request for policy change" |
| fight with artillery and tanks | "Halt military confrontation","Deescalate armed aggression" |
| Threaten with repression | "Affirm no repression","Guarantee no repression" |
| Accuse of war crimes | "Forgive","Apologize","Praise or endorse","Express accord" |
| Appeal for change in leadership | "Support the current leadership","Endorse the present administration","Back the existing leaders","Affirm confidence in leadership" |
| Conduct strike or boycott | "Continue working cooperatively","Support ongoing business" |
| Decline comment | "Provide an open statement","Give a candid response" |
| Demand material aid | "Offer defense support", "Provide military assistance" |
| Reduce or stop economic assistance | "Expand or begin economic support","Boost financial aid" |
| Make empathetic comment | "Make insensitive remark","Display no sympathy" |
| Make a visit | "Skip a visit","does not visit" |
| Increase military alert status | "Downgrade military alert level","Reduce military alertness" |
| Impose embargo, boycott, or sanctions | "Express intent to ease embargo, boycott, or sanctions","ease embargo, boycott, or sanctions" |
| Host a visit | "Cancel upcoming reception","Withdraw visiting invitations" |
| Grant diplomatic recognition | "Withdraw acknowledgment","Rescind formal relations" |

**Table 9: Examples of manually designed inverse relations for original temporal knowledge graph relations**

| Datasets | Num Entities | Num Relations | Num Events Train | Num Events Test | Num Events Val |
|---|---|---|---|---|---|
| ICEWS14 | 6,869 | 230 | 74,845 | 8,514 | 7,371 |
| ICEWS18 | 23,033 | 256 | 373,018 | 45,995 | 49545 |
| ICEWS0515 | 10,094 | 251 | 368,868 | 46,302 | 46,159 |

**Table 10: The statistics of three ICEWS datasets. Num Events Train/Test/Val denotes the number of facts in training, testing, and validation sets. Num entities/relations means the number of unique entities/relations in each dataset.**

