# OpenReview forum: "Back to the Future: Towards Explainable Temporal Reasoning with Large Language Models"
_ACM.org/TheWebConf/2024/Conference — TheWebConf24_

### Official Review · Reviewer_bcKe · 2023-11-19

**Novelty:** 6
**Technical Quality:** 5

**Review:**

Pros:
+ The paper proposed a novel knowledge graph-instructed-generation strategy and constructed a dataset called ExpTime with 26k samples containing temporal reasoning paths in the context of each sample.
+ The paper fine-tuned the LlaMA2 model and proposed a new LLM series TimeLlaMA with the ability of instruction following for explainable temporal reasoning.
+ The paper explored the ability of LLMs for explainable temporal reasoning that predicts an event's occurrence.
+ The paper proposed the method of dataset construction based on the temporal knowledge graph instruct generation.
+ The work of fine-tuning LLMs is quite new and interesting.

Cons:
- Only one section discusses how to fine-tune Llama-7b/13b models but most contents of the paper are about dataset construction.
- For each baseline, the fine-tuned methods and parameters are not mentioned in the paper (or in the Appendix).
- Temporal reasoning could be sensitive to temporal expression, but in the paper, no evidence shows the LLMs can really understand temporal expressions.
- In paper writing, it would be better that in the section of "Baselines", each baseline starts in a new line that is more readable.

**Questions:**

* Temporal reasoning is quite an interesting research topic. But I have a question about the reasoning path. Only based on the dates and some of the facts, is it reasonable that the corresponding events are inferred. For example, the example given in the paper deduces that China will host a visit of Obama. But I think a visit between two nations is a complicated issue. How can it get to the conclusion from the only two facts of TKG? The consequence might be a result of a series of events.
* Similar to the point above, in the paper, the author mentioned that they use a defined time span but didn't show what the time span is used in their configuration (a month? a year? or a time interval?). And all the examples in the paper are on the same date, which is too strict. When the temporal expressions are more complicated (e.g., including before, after, etc), how to prove the LLMs have the ability of temporal reasoning?
* Following the point above, if there are facts in a time span (or a series of facts with different time points), can the fine-tuned LLMs do temporal reasoning according to the temporal value or can LLMs really recognize the temporal order in the facts? Since the given examples in the paper do not show the ability of the fine-tuned LLMs.
* Only one section discusses how to fine-tune Llama-7b/13b models. Why don't the authors provide more details about fine-tuning LLMs?

**Reviewer Confidence:**

3: The reviewer is confident but not certain that the evaluation is correct

**Scope:**

4: The work is relevant to the Web and to the track, and is of broad interest to the community

---

### Official Review · Reviewer_mPgL · 2023-11-22

**Novelty:** 6
**Technical Quality:** 6

**Review:**

This work investigates the LLM's temporal reasoning capability based on temporal knowledge graphs and goes beyond by creating dataset for evaluation and instruction tuning and providing new pre-trained temporal LLMs. The paper is comprehensive, very well conducted and written.

Perhaps the only, minor issue is that RQ1 asks about comparison with traditional methods, but the most "traditional" one is an advanced encoder-decoder model, BART, which seems not very traditional. Would be interesting to know more about how other earlier, traditional methods. This is, however, optional.

Other minor comments:
1 It is not sure whether the process of letting ChatGPT evaluate F_prime and polish and revise the prompt is a “chain-of-thought” (COT) process.
2. Would be interested to know what further temporal reasoning tasks are challenging for the future as stated in the last sentence in the conclusion.
3. Golden-standard -> gold standard

**Questions:**

Please see the review comments.

**Reviewer Confidence:**

3: The reviewer is confident but not certain that the evaluation is correct

**Scope:**

4: The work is relevant to the Web and to the track, and is of broad interest to the community

---

### Official Review · Reviewer_5R21 · 2023-11-22

**Novelty:** 6
**Technical Quality:** 6

**Review:**

The paper focuses on the task of "explainable temporal reasoning": predicting events' occurrence at future timestamps based on multiple-event contexts and providing clear explanations for the predictions.
To support this, the authors present an instruction-tuning dataset, ExpTime, derived from previous temporal knowledge graph datasets by using explainable TKG reasoning models (TimeTraveler and TLogic) and LLMs (ChatGPT) through a procedure (“knowledge graph-instructed generation strategy”), described in the paper. Such a dataset contains, given an input document and query, both predictions and corresponding explanations.
Using ExpTime the authors train an open-source LLM series, TimeLlaMA, which is later shown to outperform other LLMs, demonstrating state-of-the-art results in temporal prediction and explanation generation.

The paper is well written and, despite the inherent complexity of the detailed procedures, relatively easy to read.
The topic and methodology of the paper is very relevant, as they focus on explainable event prediction, and make use of knowledge graph to inject knowledge in LLMs (in this case, by exploiting KGs and inference techniques to programmatically build a database with which to fine-tune LLMs).
The paper argues convincingly in favor of its main arguments:
- that the procedure detailed in the paper builds high quality datasets for the task of explainable temporal reasoning.
- that the ensuing databases can be used to fine tune LLMs (on the task of explainable temporal reasoning) to noticeably improve performance

Minor issues:
At line 432, t_j occurs but is not present in the previous portion of the text, what is it?
At line 1199, “Pprompt”

**Questions:**

In Figure 1 the flags of south Korea and Japan are partially indistinguishable from the white background.
Moreover, given that “We also add the quadruples along the reasoning path 𝑃𝑎𝑖 to the context set” shouldn’t the figure contain also an arrow from “reasoning paths” to “context quadruples”?

**Reviewer Confidence:**

2: The reviewer is willing to defend the evaluation, but it is likely that the reviewer did not understand parts of the paper

**Scope:**

4: The work is relevant to the Web and to the track, and is of broad interest to the community

---

### Official Review · Reviewer_7h9j · 2023-11-28

**Novelty:** 4
**Technical Quality:** 2

**Review:**

The submission proposes to use ChatGPT (and other LLMs) for generating explanations for sequences of events. The idea is interesting, but is very poorly presented, with lots of imprecisions and misleading statements.

Using ExpTime for a dataset is not a good idea - normally, ExpTime refers to the class of problem that can be solved in exponential time.

It appears that "instruction tuning" (l 131) has a special meaning, which is not explained.

What is the meaning of the \sim operator in l 326?

It's not very clear why (1) is called a "reasoning chain" - it looks more like A & B -> C, not A -> B -> C. The chain A -> B -> C is indeed illustrated in (2), but formula (1) is more about the following: IF "it is cold" AND "an atmospheric front is approaching" THEN "it is likely to snow", which does not look like (2).

The explanation template in l 455-459 makes no sense - it first claims something, and then says "and therefore that something holds". Of course, if you claim "A", then you get "therefore A" for free - it's called a tautology!

 What is a "Polish prompt" mentioned in l 520? Does it have anything to do with Poland?

Inverse (or converse) is not the right term for the relations the authors mean - it's really confusing. The converse of IF "it is snowing" THEN "it is cold" is IF "it is cold" THEN "it is snowing". We use "conversely" to change the direction of the implication. In the same way, we can talk about the "inverse of a function": e.g., the square root is the inverse of the x^2 function. What the authors appear to mean is the complement - "X had rain on day Y" is the complement of "X had no rain on day Y".




l 30: 26k of what?

l 31-32 & 917: too many dashes; l 155, on the other hand, has too few dashes

l 103: why "focused", in the past?

l 107: "it is still unclear the potential" is broken

l 146: what is "built"? Does it refer to the dataset?

l 193: encapsulate is not the right word; summarize? recapitulate?

l 314: it's a bit too late to introduce the LLM as an abbreviation

l 341-342: it's not very clear how the GIG abbreviation is obtained (normally, capital letters are used to indicate the parts that form an abbreviation)

l 432: it's a bit too complex for a simple interval expression (ti < tq < tj)

l 581: what is 0515? is ICEWS not missing there?

l 603 & 636 & 693: fine*-*tuning

l 669: A*n* LLM

l 670: what is 7b?

**Questions:**

How do the authors deal with hallucinating LLMs?

**Ethics Review Description:**

-

**Reviewer Confidence:**

3: The reviewer is confident but not certain that the evaluation is correct

**Scope:**

3: The work is somewhat relevant to the Web and to the track, and is of narrow interest to a sub-community

---

### Decision · Program_Chairs · 2024-01-22

**Decision:**

Accept

**Comment:**

* There is generally no disagreement. Authors are encouraged to incorporate results of reviewer discussion in the event of an accepted paper.